# Photobiomodulation for pain management during placement of the copper T 380 intrauterine device: Protocol for a randomized, double-blind controlled trial

Anna Carolina Nunes Ferraz[1,2☺¤a¤b], Maria Aparecida Traverzim[1,2☺¤a¤b],
Sergio Makabe[1,2☺¤a¤b], Gláucia Gonçalves Abud Machado[1☺¤a],
Sandra Kalil Bussadori[1,3☺¤a¤c], Raquel Agnelli Mesquita-Ferrari[1,3☺¤a¤c],
Cinthya Cosme Gutierrez Duran[1☺¤a], Lara Jansiski Motta[1,3☺¤a],
Kristianne Porta Santos Fernandes[1☺¤a], Anna Carolina Ratto Tempestini Horliana[1☺¤a*]

**1** Postgraduate Program in Biophotonics-Medicine, Universidade Nove de Julho (UNINOVE), São Paulo, Brazil, **2** Mandaqui Hospital Complex, São Paulo, Brazil, **3** Postgraduate Program in Rehabilitation Sciences, Universidade Nove de Julho (UNINOVE), São Paulo, Brazil

☺ These authors contributed equally to this work.
¤a Current address: Postgraduate Program in Biophotonics-Medicine, 235/249 Vergueiro Street, 01525−000 São Paulo, SP, Brazil
¤b Current address:Rua Voluntários da Pátria, 4301, Santana, Zona Norte de São Paulo, SP, Brazil
¤c Current address:Postgraduate Program in Biophotonics-Medicine, 235/249 Vergueiro Street, 01525−000 São Paulo, SP, Brazil
* annacrth@gmail.com

## Abstract

### Introduction

Unplanned pregnancy remains a critical public health issue in Brazil that affects up to 65% of women in some regions and increases the incidence of both unsafe abortions and maternal mortality. Although the copper intrauterine device (IUD) is a highly effective, long-acting contraceptive method, its rate of use remains low in Brazil (only 4.4% of women of reproductive age). One of the primary barriers is the pain experienced during placement of the device, which discourages adherence. Although multiple social and cultural determinants influence IUD uptake, insertion-related pain constitutes a modifiable, procedure-specific factor that may directly affect both pain perception and anticipatory discomfort. Conventional pain management strategies, such as anti-inflammatories and local anesthetics, have produced inconsistent results. Photobiomodulation (PBM) is a non-invasive therapy with anti-inflammatory and analgesic properties that has shown promise in managing pelvic pain in contexts such as labor.

### Methods and analysis

The proposed randomized, double-blind clinical trial will investigate the effectiveness of PBM at reducing pain during the placement of the copper T 380 IUD. Seventy-two

**Data availability statement:** Deidentified research data will be made publicly available when the study is completed and published.

**Funding:** The author(s) received no specific funding for this work.

**Competing interests:** The authors have declared that no competing interests exist.

women will be randomly allocated to either an active PBM group (n = 36) or sham PBM control group (n = 36). In all participants, the standard IUD placement protocol will follow the guidelines of the Brazilian Ministry of Health. Pain will be assessed using the Visual Analog Scale (VAS) at multiple time points: during each step of insertion (Pozzi clamp, hysterometry, IUD placement) as well as 5 min, 15 min, 24 hours, and 48 hours post-procedure. The secondary outcomes will be analgesic use, anxiety levels (GAD-7), quality of life (WHOQOL-100), satisfaction with the procedure, and adverse effects. Ethics and dissemination. This clinical trial received approval from the Research Ethics Committee of the Mandaqui Hospital Complex (process: 7.367.867) and was registered in ClinicalTrials.gov (NCT06984796). Statistical analyses will be conducted using SPSS v24.0, with a significance level of 5%. Either parametric or nonparametric tests will be applied, depending on the data distribution. Kaplan-Meier curves will be used to assess time to pain resolution. The results will be disseminated through peer-reviewed publications.

## Introduction

Unplanned pregnancy affects up to 65% of women in some areas of Brazil, increasing the risk of unsafe abortions and inadequate prenatal care, which significantly contribute to maternal mortality. Ensuring access to long-acting contraceptive methods, such as the copper intrauterine device (IUD), is crucial in reproductive planning [1,2]. Despite its effectiveness, only 4.4% of women use an IUD, with many opting for birth control pills or sterilization [3]. The IUD is suitable for most women of reproductive age, including adolescents, and is also indicated for emergency contraception and controlling heavy uterine bleeding [4].

Despite these benefits, pain during placement causes fear and reluctance among patients and constitutes a significant barrier to IUD adoption [5,6]. While broader social and cultural determinants influence IUD uptake, insertion-related pain is an immediate, modifiable procedural factor that may directly affect both pain perception and anticipatory fear. Visceral pain arises from cervical dilation, with afferent nerves transmitting impulses via the T10-L1 dermatomes [7], while somatic pain results from pelvic floor distension, involving the S2-S4 dermatomes [8]. Evidence of adequate pain control remains inconclusive, as studies involving the use of non-steroidal anti-inflammatory drugs (NSAIDs), local anesthetics, and cervical preparation methods have demonstrated limited effectiveness [2,5–7]. A Cochrane review found that 2% lidocaine gel, misoprostol, and most NSAIDs failed to reduce pain during IUD placement, with only some formulations of lidocaine, tramadol, and naproxen having effects in specific groups [6].

Given the lack of effective pharmacological interventions, alternative approaches have been explored. Paravertebral blockade between T10 and S4 has shown promise for analgesia during labor [2,5,9], while transcutaneous electrical nerve stimulation (TENS) has been effective at controlling pelvic pain [10,11]. Photobiomodulation (PBM) has demonstrated analgesic effects for lower back pain [12,13] and labor pain

[9], suggesting the potential to reduce discomfort during IUD placement [14]. As the Brazilian Ministry of Health does not currently recommend analgesic measures for IUD placement, further studies are needed to assess the effectiveness of PBM at improving patient comfort and increasing adherence to this contraceptive method, ultimately benefiting public health [4]. Our experimental hypothesis is that the use of PBM can reduce pain perception during the placement of the T 380 copper IUD for contraception.

Therefore, the aim of this study is to propose a protocol for the assessment of whether preventive photobiomodulation (PBM) alters pain perception during the placement of the T 380 copper intrauterine device (IUD) for contraception in women without comorbidities.

## Materials and methods

A single-center, randomized, double-blind, controlled clinical trial with two parallel groups, designed as a superiority trial, will be conducted following the schedule of enrollment (Fig 1) and SPIRIT Statement (https://www.spirit-statement.org/) (S1 File).

The study protocol (original language S2 File; in English S3 File) received approval from the Research Ethics Committee of the Mandaqui Hospital Complex in São Paulo, Brazil (7.367.867). The original document (S4 File) and a translation (S5 File) were presented Any unforeseen events or modifications during the study will be reported to the ethics committee and disclosed in future publications. After the main investigator provides a verbal and written explanation of the study, individuals who agree to participate will sign an informed consent form. Participants who wish to receive the study results will provide their email addresses and the full article will be shared upon publication. The participants will be involved in the development of the study design and the execution of the trial but not involved in the data analysis or reporting.

Treatment will be performed at the Mandaqui Hospital Complex in the northern region of São Paulo, Brazil. The main investigator is a gynecologist with more than 10 years of experience and will perform the procedures. According to the ClinicalTrials.gov record (NCT06984796) (S6 File), the study is currently listed as "Not yet recruiting", with an anticipated starting date of March 30, 2026, and a primary completion date of May 30, 2027. The study was registered in ClinicalTrials.gov (NCT 06984796) and is in accordance with the SPIRIT 2025 Guidelines Checklist Statement.

### Calibration/Training

A single examiner will assess five women who will not be included in the study. These women will undergo clinical pain assessments identical to those proposed for the trial.

The main investigator will perform all IUD placement procedures and postprocedural assessment. This investigator will also undergo specific training for the following outcomes: anxiety, using the Generalized Anxiety Disorder 7 (GAD-7) questionnaire; quality of life, using the WHOQOL-Pain instrument; and patient satisfaction, using a questionnaire based on the method proposed by Lopes *et al.* (2015) [6].

### Sample size calculation

The total sample size will be 72 participants. This number was calculated to provide an 80% power ($\alpha = 0.05$) with an effect size of 0.14. The number of patients per group was determined using a sample size calculation with G*Power 3.1.9.7. The sample was calculated for two groups, and 8 measurements will be performed: at baseline, during the IUD insertion phases (Pozzi, hysterometry, and IUD placement), at 5 and 15 minutes after placement, and at 24 and 48 hours after the procedure.

### Sample description

Women of reproductive age will be selected from those referred to the hospital for contraception from primary care units in the Northern Zone of the city of São Paulo or the Gynecological Emergency Department of the Mandaqui Hospital Complex.

| | Enrolment | Allocation | Post-allocation | | | | | | Close-out |
|---|---|---|---|---|---|---|---|---|---|
| **STUDY PERIOD** | | | | | | | | | |
| TIMEPOINT | Anamnesis | 0 | Cervical Clamping | Hysterometry | Insertion | 5 min | 15 min | 24 h | 48 h |
| **ENROLMENT:** | | | | | | | | | |
| Eligibility screen | x | | | | | | | | |
| Informed consent | x | | | | | | | | |
| Allocation | | x | | | | | | | |
| **INTERVENTIONS:** | | | | | | | | | |
| Insertion of the T 380 copper IUD | | | x | | | | | | |
| PBM simulation | | | x | | | | | | |
| PBM | | | x | | | | | | |
| **ASSESSMENTS:** | | | | | | | | | |
| VAS assessment | x | | x | x | x | x | x | x | x |
| Satisfaction questionnaire | x | | | | | | x | | x |
| Anxiety questionnaires | x | | | | | | x | | |
| Analgesic count (number and time point) | | | | | | | | | x |
| Time to take 1st analgesic | | | | | | | | | x |
| Quality of life | x | | | | | | | | x |
| Success of treatment | | | | | | | | | |
| Collateral and adverse events | | | | | | | | | |

**Fig 1. Template illustrating recommended content for schedule of enrollment, interventions, and assessments.**

## Inclusion and exclusion criteria

The study will include women aged 18–50, with no restrictions on race or socioeconomic status. Both nulliparous and multiparous women will be considered eligible, provided they meet other criteria established to ensure the safety and appropriateness of the IUD placement procedure.

Conversely, women with suspected or confirmed pregnancy, a history of diagnosed chronic pain, or active pelvic infection will be excluded from the study. Women who took pain medication within 12 hours before the procedure will be excluded to prevent interference with pain perception. Women with known contraindications for IUD placement, such as significant uterine cavity distortion, active pelvic inflammatory disease, or Wilson's disease, will be deemed ineligible [4]. Other exclusion criteria include copper allergy, unexplained abnormal uterine bleeding [4], a history of photosensitivity, or any condition affecting the lumbar region, such as active neoplasms, established osteomyelitis, or pre-existing deep tissue injuries with necrosis or infection. The participants will not be permitted to participate in any other clinical study during the trial.

## Study discontinuation criteria

There are no plans to interrupt or modify the allocated interventions for any participant in the trial, as the control group will receive the standard treatment recommended by the Brazilian Ministry of Health, a systematic Cochrane review [6], and the manufacturer's guidelines for IUD placement. There is no evidence of an effective analgesic treatment for IUD placement [6]. The intervention in the experimental group is a procedure that does not cause adverse effects or complications, with previous reports indicating analgesic effects during labor [9]. As the study poses no risk to the participants, there is no criterion for study discontinuation. Given the very short duration of participant involvement in the study, no specific strategies for improving or monitoring adherence will be required.

## Recruitment

Recruitment will be based on the voluntary demand of women referred to the hospital for contraception from primary care units in the Northern Zone of the city of São Paulo or the Gynecological Emergency Department of the Mandaqui Hospital Complex.

## Collection of patient history and risk factors

The patient history will be collected from all participants using a questionnaire. In addition to general demographic data (age, sex, marital status, occupation, educational level, and income), the medical history will cover chief complaints, current disease status, past medical history, medications in use, smoking, alcohol consumption, the use of systemic antibiotic therapy in the previous 12 months, hospitalization or infection (including skin infections) in the previous 12 months, previous IUD use, and preferred contraceptive methods. This questionnaire will enable the correlation of clinical, demographic, and epidemiological risk factors with the study outcomes in this population.

## Allocation (Sequence generation, allocation concealment mechanism, implementation)

A random sequence generator available online (https://www.sealedenvelope.com/) will be used to select the block randomization option with two treatments at a 1:1 ratio, resulting in equal numbers of patients in each group (n = 36 per group). As the total sample size is 72 patients, 12 blocks of six patients each will be established. Opaque envelopes will be sequentially numbered (1–72) and will contain the randomized group allocation. These envelopes will be sealed and stored securely until the IUD is placed. Only the designated researcher responsible for treatment will be aware of the assigned intervention. After opening, the envelope will be resealed and stored with the patient's medical record, remaining concealed until the final statistical analysis. An individual not directly involved in the study will generate the random sequence and prepare the envelopes.

All participants will undergo the same IUD placement protocol performed by the same healthcare provider, as described in the "IUD Placement Protocol" section. Immediately after placement, the researcher in charge of applying PBM will retrieve and open an envelope, maintain the order of the remaining envelopes, and administer the assigned treatment (active or sham PBM). Only this researcher will be aware of the allocation.

The main investigator will oversee data storage (digital platforms) and organization. The statistician will be responsible for data analysis and will remain blinded to the interventions. The envelopes will also store participant records (patient history questionnaire, demographic data, risk factors, and statement of informed consent) and will be securely archived.

## Blinding (Masking)

Only the researcher in charge of administering the treatment (who opens the randomization envelopes) will be aware of the assigned intervention for each participant. The participants will be blinded because of the visual similarity of the treatments (LED on versus LED off). The researcher collecting the outcome data will also be blinded to the interventions and will not be present in the room during treatment. Group allocation will be disclosed only after the statistical analysis is completed, ensuring that the data collector and statistician remain blinded. Unblinding during the trial is not planned, as no adverse events are expected with PBM use.

## IUD placement protocol according to the recommendations of the ministry of health

Before the procedure, the steps will be explained to the patient again to reduce anxiety and promote relaxation during IUD placement. A two-hand pelvic examination will be performed to assess the size, position, and mobility of the uterus. Infection prevention measures will be implemented, including the use of sterile gloves and cervical cleansing with a chlorhexidine-based antiseptic. The anterior lip of the cervix will be gently grasped with Pozzi forceps to ensure stabilization during placement. A hysterometer will be introduced slowly and carefully to determine uterine depth and angulation, thus minimizing the risk of perforation. During IUD preparation, all instruments and gloves will be maintained sterile, and the device will be assembled according to the manufacturer's instructions, with the arms kept horizontal during insertion. The retractable technique will be used for placement. The insertion tube will be advanced to the uterine fundus. The tube will then be partially withdrawn while keeping the plunger fixed, thereby releasing the arms of the IUD. After a few seconds, the plunger and insertion tube will be removed. The IUD strings will be trimmed to 2–3 cm from the cervix. The patient will remain lying down for approximately 15 minutes to reduce discomfort. Following placement, the patient will be monitored for well-being and possible vasovagal reactions (e.g., sweating, nausea, or fainting), which are rare and self-limiting. The length of the IUD strings relative to the cervix will be documented in the medical record. The patient will be instructed to return for a follow-up visit in 30 days to verify IUD positioning and adaptation.

## Group composition

The 72 participants will be allocated to the control and experimental groups as follows (Figs 1 and 2):

- Control group – PBM simulation (n = 36 participants): All participants in this group will undergo the conventional IUD placement procedure, as described above. They will receive sham PBM and be treated the same as the experimental group. The researcher in charge of the PBM application will simulate irradiation by positioning the device in the exact location as in the experimental group, but the equipment will be switched off. To prevent the participants from identifying the group to which they belong, the activation sound of the device (beep) will be pre-recorded and played during the sham application.

- Experimental group – active PBM (n = 36 participants): All participants will undergo the same IUD placement procedure. For irradiation, an LED panel from Sportlux® (Brazil, SP) (Fig 3) will be used. The equipment description, dosimetric parameters, and number of PBM applications are detailed in Table 1.

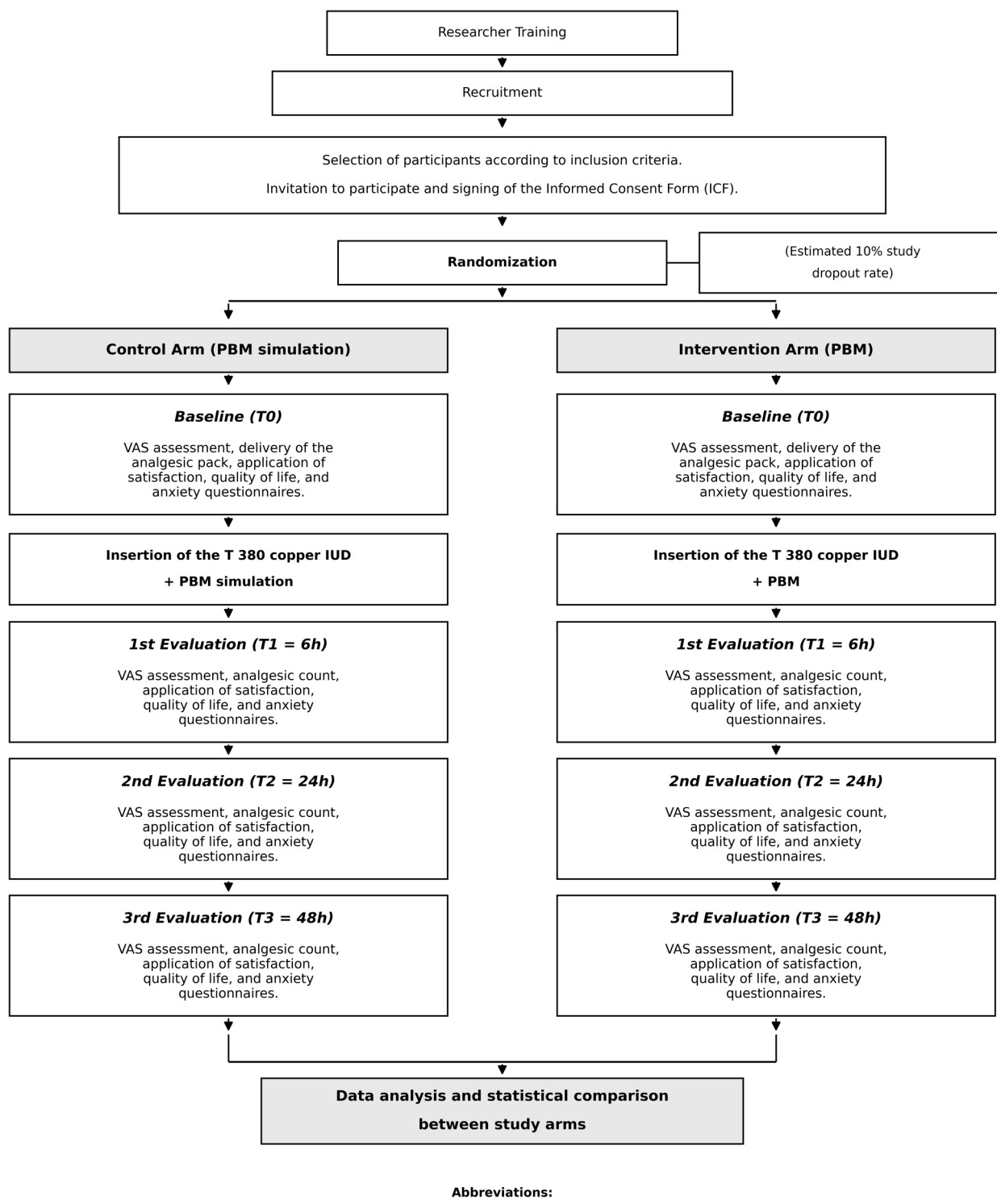

**Abbreviations:**

ICF = Informed Consent Form;  IUD = Intrauterine Device;

PBM = Photobiomodulation;  VAS = Visual Analog Scale.

**Fig 2.  Flowchart of study.**

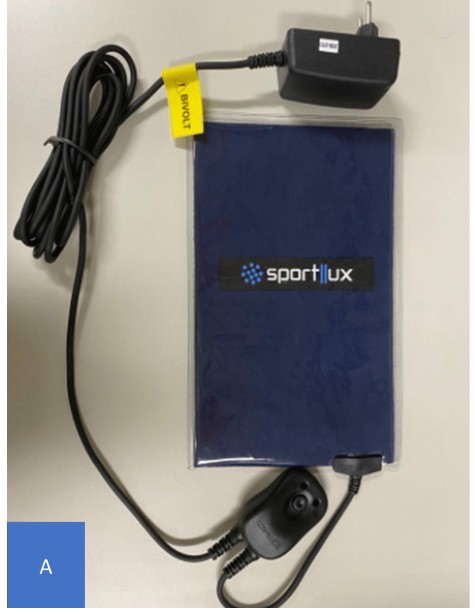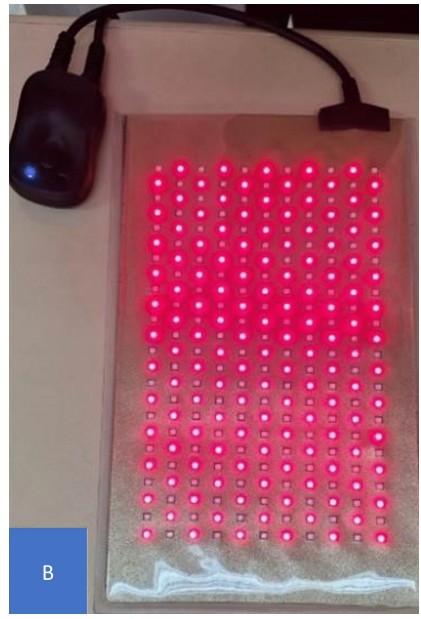

**Fig 3. Sportlux® LED panel used for preventive irradiation during IUD placement.** Legend: Source: Created by the authors. Figure A: Panel switched off. Figure B: Panel switched on.

**Table 1. Dosimetric Parameters Used for Preventive PBM.**

| Technical Parameters | Values |
|---|---|
| Light source | LED _ |
| Application technique | Contact |
| Wavelength | 132 LEDs at 660 nm/ 132 LEDs at 850 nm |
| Spectral band | 20 nm |
| Beam area on target | 0.5 cm² |
| Irradiation time | 20 min |
| Average power per LED | 8 mW |
| Irradiance | 16 mW/cm² |
| Application time | 10 minutes |
| Energy per LED | 4.8 J |
| Radiant exposure | 9.6 J/cm² |
| Light emission angle | 120° |

nm- nanometro, W- Watt, J- Joule.

## Outcome variables

### Primary outcome.

- Pain – Assessed using the Visual Analog Scale (VAS) at baseline, during the IUD placement phases (Pozzi, hysterometry, and IUD placement), 5 and 15 minutes after placement as well as 24 and 48 hours after the procedure.

**Secondary outcomes.**

- Pain description – Free narrative report

- Number of analgesics taken – Paracetamol intake recorded from baseline to 48 hours after T 380 copper IUD placement.

- Anxiety – Assessed using the Generalized Anxiety Disorder 7 (GAD-7) questionnaire [15] 15 minutes after IUD placement.

- Quality of life – Measured using the WHOQOL-Pain instrument [16] from baseline to 48 hours after placement.

- Patient satisfaction – Assessed through close-ended questions [6] 15 minutes after placement.

- Time required for relief of abdominal discomfort (menstrual cramps) – Measured in days from baseline.

- Verification of successful placement – Investigated dichotomously (success/failure) from baseline to 48 hours after placement.

- Adverse effects – Including uterine perforation, IUD displacement, abdominal pain, increased vaginal bleeding, and allergic reactions. An open-ended question will be used for the participants to report adverse effects, followed by a list of potential impacts to help them recall any unreported symptoms.

- Side effects – including cramps, mild pain, light bleeding, and tongue numbness. An open-ended question will be asked first, followed by a list of potential side effects to improve recall.

*Primary outcome – Pain - Visual Analog Scale (VAS)*: Pain will be assessed using the Visual Analog Scale (VAS), as illustrated below. One end of the scale is marked "0", corresponding to the absence of pain, and the other end is marked "10", indicating unbearable pain. The same scale will be used for all participants. The same operator will always provide instructions for marking the scale. Pain scores will be recorded, and this assessment will be performed at all study visits, including during the IUD insertion phases (Pozzi, hysterometry, and IUD placement), at 5 and 15 minutes after placement, and at 24 and 48 hours after the procedure. The participants will be asked about spontaneous pain at three distinct moments during placement: pain during clamping with the Pozzi forceps, pain during hysterometry (uterine length measurement), and pain during the actual IUD placement. All participants will receive daily phone call reminders to ensure accurate recording of their outcomes [17].

- Cervical clamping: Gently clamp the anterior lip of the cervix using Pozzi forceps to stabilize it during placement.

- Hysterometry: Slowly and carefully insert the hysterometer to determine uterine depth and angulation, minimizing the risk of uterine perforation.

- Placement of IUD in the uterine cavity

 *Secondary outcome – Description of pain – Free Narrative*: Immediately after IUD placement, the patients will be asked to describe their experience with the procedure. The responses will be recorded verbatim to enable qualitative analysis.

 *Secondary outcome – Quantity of analgesics (amount taken by the patient will be recorded at 24 and 48 h after the T 380 copper IUD placement*: Analgesic (paracetamol) consumption will be assessed as proposed by Bauer et al. (2013; [18]). At the beginning of the study, each participant will receive a blister pack of paracetamol, which is a pure analgesic [19]. The blister pack must be kept until the end of the study, and the analgesic will be used only for pain. At the end of the study, the number of remaining tablets will be quantified as an additional measure of pain. The analgesic will be paracetamol 500 mg to be taken only when necessary at a dosage of one tablet every six hours. The participants will be asked to record in writing the number of tablets taken, along with the date and time of consumption. A monitoring procedure will

be implemented to ensure adherence: the participants will be required to bring their blister pack to follow-up visits for verification. Medication use will be tracked from baseline through 48 hours post-placement.

*Secondary outcome – Anxiety assessment*: Anxiety levels in patients undergoing T 380 copper IUD placement for contraception will be assessed using the seven-item Generalized Anxiety Disorder (GAD-7) questionnaire administered at baseline and 15 minutes after the procedure [15]. The GAD-7 is a self-report instrument designed to assess generalized anxiety symptoms based on the DSM-IV diagnostic criteria. The questionnaire consists of seven items measuring the frequency of anxiety-related issues over the previous two weeks, using a 4-point Likert scale: 0 = Never, 1 = Several days, 2 = More than half the days, 3 = Almost every day. The GAD-7 can be administered in clinical and research settings, both in person and online, and takes approximately five minutes to complete. The total score ranges from 0 to 21 points, with higher scores indicating greater severity of anxiety symptoms. Classification is divided into four categories: scores from 0 to 4 indicate no anxiety, 5–9 correspond to mild anxiety, 10–14 correspond to moderate anxiety, and 15–21 indicate severe anxiety. The GAD-7 has strong validity and reliability and is widely used in clinical and non-clinical populations. Depending on the application context, it can be analyzed using a unifactorial or bifactorial model (somatic and cognitive-emotional items).

*Secondary outcome – Assessment of quality of life using the WHOQOL-Pain instrument at baseline and 24 hours after the procedure*: The WHOQOL-Pain is an additional module of the WHOQOL-100 specifically developed to assess quality of life in individuals living with chronic pain. Following a preliminary study, ten new facets related to pain and discomfort were identified, four of which were selected for the final version of the instrument: Pain Relief; Anger and Frustration; Vulnerability, Fear and Concern; and Uncertainty [16]. Each facet comprises four questions that address different dimensions of the pain experience. The responses are recorded on a five-point Likert scale to measure intensity, capacity, evaluation, and frequency. The scores are converted to a 0–100 scale, enabling the quantitative interpretation of pain levels and the impact on quality of life. The WHOQOL-Pain items are not integrated into the WHOQOL-100 and are applied separately at the end of the main questionnaire. This structure facilitates the individualized analysis of pain-related facets, enabling a specific, detailed assessment of physical chronic pain experiences.

Patient satisfaction with T 380 copper IUD placement for contraception will be assessed 15 minutes after the procedure using guiding questions [6]. The assessment will include whether the placement experience was unpleasant, whether the patient would undergo the procedure again, and whether she would recommend it to a friend.

*Secondary outcome – Time required to relieve abdominal discomfort following PBM*: The time required to relieve abdominal discomfort following PBM will be recorded in hours, measured from the onset of cramps until complete pain resolution.

*Secondary outcome – Verification for IUD placement success*: IUD placement will be assessed as a binary outcome (success/failure).

*Secondary outcome – Adverse effects (uterine perforation, IUD displacement, abdominal pain, increased vaginal bleeding, and allergy)*: An open-ended question will be asked to elicit the patient's free responses regarding adverse effects. The responses will then be listed nominally so that the patient may recall any impact that she may have forgotten to report.

*Secondary outcome – Side effects (cramps, mild pain, mild bleeding, tongue anesthesia)*: An open-ended question will be asked to enable the patient to freely describe any side effects. Effects will then be listed by name to help the patient recall any impact that she may have forgotten to report.

### Ethical aspects

This study ensures compliance with applicable ethical and regulatory guidelines. No modifications to the research protocol are planned; however, if any arise, they will be promptly communicated to all relevant parties, including investigators, the ethics committee (via an amendment submission), participants, and clinical trial registries. A

trained researcher will obtain informed consent after providing the participants with detailed information about the objectives, procedures, risks, and benefits of the study, using a consent form written in clear, accessible language. No biological materials will be collected or stored. The collection, sharing, and storage of the participants' personal information will follow strict data protection protocols to ensure confidentiality before, during, and after the trial. All data will be anonymized and securely stored to prevent unauthorized access. There are no financial or other conflicts of interest. Only the main investigator will have access to the final dataset. The researcher in charge is committed to disseminating the survey results to the participants, healthcare professionals, and the public through publications and presentations at conferences while respecting any previously agreed-upon publication restrictions.

## Statistical analysis

Statistical analysis will be performed using the Statistical Package for the Social Sciences (SPSS) version 24.0 or equivalent software package. The significance level will be set at 5% ($p < 0.05$) for all tests. Data normality will be determined using the Shapiro-Wilk test. Continuous variables will be reported as either the mean and standard deviation or the median and interquartile range, depending on the distribution of the data. Categorical variables will be expressed as absolute and relative frequencies. For comparisons between the experimental group (active PBM) and control group (sham PBM), the Mann-Whitney test will be used for continuous variables with non-normal distribution. When the expected minimum value is less than 5, categorical variables will be compared using either the chi-square test or Fisher's exact test. The analysis of pain-related variables (intensity measured by the Visual Analog Scale [VAS]) at different time points (baseline, during placement, 5 and 15 minutes, and 24 hours after the procedure) will be conducted using the Friedman test. As the primary outcome will be measured repeatedly over time, the analysis will account for the within-subject correlation inherent to this repeated-measures design.

For the measures of anxiety (GAD-7), quality of life (WHOQOL-100), and satisfaction with the procedure, the Mann-Whitney test will be used to compare groups across assessment time points. The use of post-procedural analgesics will be compared between groups using repeated-measures analysis of variance (ANOVA). No interim analyses or stopping guidelines are planned due to the short duration of the trial and low risk.

Missing data will be assessed for frequency and pattern. In the event of missing values, the analyses will be performed using the available data, and the extent of missingness will be reported. If necessary, appropriate statistical methods will be applied depending on the nature and proportion of the missing data.

## Data management and quality control

All data will be collected using standardized forms and recorded by trained researchers. The data will be entered into a secure electronic database and checked for completeness and consistency. Periodic data verification will be performed to identify inconsistencies or missing values, which will be reviewed and corrected against the original records. Access to the database will be restricted to authorized members of the research team. All data will be anonymized prior to analysis to ensure participant confidentiality.

## Supporting information

**S1 File. This is the S1_File Title Spirit.** This is the S1 File legend; there is no legend.
(DOCX)

**S2 File. This is the S2_File Title Appendix IRB approval in the original language.** This is the S2 File legend; there is no legend.
(DOCX)

**S3 File. This is the S3 File Title Appendix IRB approval in English.** This is the S5 Fig legend; there is no legend.
(DOCX)

**S4 File. This is the S4_File Title Appendix study protocol IRB original language.** This is the S3 File legend; there is no legend.
(PDF)

**S5 File. This is the S5_File Title Appendix study protocol IRB in English.** This is the S4 File legend; there is no legend.
(PDF)

**S6 File. This is the S6_File Title clinical_trials.** This is the S6 File legend; there is no legend.
(PDF)

## Author contributions

**Conceptualization:** Anna Carolina Nunes Ferraz, Maria Aparecida Traverzim, Sergio Makabe, Sandra Kalil Bussadori, Raquel Agnelli Mesquita-Ferrari, Cinthya Cosme Gutierrez Duran, Kristianne Porta Santos Fernandes, Anna Carolina Ratto Tempestini Horliana.

**Data curation:** Maria Aparecida Traverzim.

**Formal analysis:** Gláucia Gonçalves Abud Machado, Sandra Kalil Bussadori, Cinthya Cosme Gutierrez Duran, Lara Jansiski Motta.

**Funding acquisition:** Anna Carolina Nunes Ferraz, Maria Aparecida Traverzim, Sergio Makabe.

**Investigation:** Gláucia Gonçalves Abud Machado, Kristianne Porta Santos Fernandes, Anna Carolina Ratto Tempestini Horliana.

**Methodology:** Anna Carolina Nunes Ferraz, Sandra Kalil Bussadori, Raquel Agnelli Mesquita-Ferrari, Cinthya Cosme Gutierrez Duran, Lara Jansiski Motta, Kristianne Porta Santos Fernandes, Anna Carolina Ratto Tempestini Horliana.

**Project administration:** Maria Aparecida Traverzim, Sergio Makabe.

**Supervision:** Sergio Makabe, Sandra Kalil Bussadori, Raquel Agnelli Mesquita-Ferrari, Cinthya Cosme Gutierrez Duran, Lara Jansiski Motta, Kristianne Porta Santos Fernandes, Anna Carolina Ratto Tempestini Horliana.

**Validation:** Raquel Agnelli Mesquita-Ferrari, Kristianne Porta Santos Fernandes, Anna Carolina Ratto Tempestini Horliana.

**Visualization:** Gláucia Gonçalves Abud Machado.

**Writing – original draft:** Anna Carolina Nunes Ferraz, Maria Aparecida Traverzim, Sergio Makabe, Gláucia Gonçalves Abud Machado.

**Writing – review & editing:** Sandra Kalil Bussadori, Raquel Agnelli Mesquita-Ferrari, Cinthya Cosme Gutierrez Duran, Lara Jansiski Motta, Kristianne Porta Santos Fernandes, Anna Carolina Ratto Tempestini Horliana.

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
