## [Decision Letter · Decision Letter 0]

10 Mar 2026

PONE-D-26-05815Photobiomodulation for pain management during copper T 380 intrauterine device insertion: protocol for a randomised, double-blind controlled trialPLOS One

Dear Dr. Horliana,

Thank you for submitting your manuscript to PLOS ONE. After careful consideration, we feel that it has merit but does not fully meet PLOS ONE’s publication criteria as it currently stands. Therefore, we invite you to submit a revised manuscript that addresses the points raised during the review process.

Also, the protocol manuscript requires native-English-language editor services for proofreading and copyediting.

We look forward to receiving your revised manuscript.

Kind regards,

Syed Khurram Azmat, PhD, MPH, MD

Academic Editor

PLOS One

2. We note that the original protocol that you have uploaded as a Supporting Information file contains an institutional logo. As this logo is likely copyrighted, we ask that you please remove it from this file and upload an updated version upon resubmission.

Reviewers' comments:

Reviewer's Responses to Questions

**Comments to the Author**

1. Does the manuscript provide a valid rationale for the proposed study, with clearly identified and justified research questions?

Reviewer #1: Yes

2. Is the protocol technically sound and planned in a manner that will lead to a meaningful outcome and allow testing the stated hypotheses?

Reviewer #1: Yes

3. Is the methodology feasible and described in sufficient detail to allow the work to be replicable?

Reviewer #1: Yes

4. Have the authors described where all data underlying the findings will be made available when the study is complete?

Reviewer #1: Yes

5. Is the manuscript presented in an intelligible fashion and written in standard English?

Reviewer #1: Yes

6. Review Comments to the Author

You may also provide optional suggestions and comments to authors that they might find helpful in planning their study.

Reviewer #1: My comments are primarily evaluation of the statistical design, data analysis and data management.

The protocol is well written from a study design and analysis perspective. The sample size is derived appropriately with power considerations. The effect size of 0.14 is reasonable. However, the exact endpoint (presumably VAS if that is the primary) should be given and the appropriate measure being reported since this appears to be a repeated measures design. The Friedman test given in the ‘Statistical Analysis’ section is certainly appropriate.

Figure 2 appears to be complete for timeline and assessments.

There are some issues, easily addressed by the investigators.

The study needs a data management and data quality control section. These are missing and one does not see them in the supplemental materials. Also are there going to be any issues with missing data and , if so, how will that be handled?

7. PLOS authors have the option to publish the peer review history of their article (what does this mean?). If published, this will include your full peer review and any attached files.

Reviewer #1: No

---

## [Author Response · Author response to Decision Letter 1]

27 Mar 2026

Reviewers' comments:

Reviewer's Responses to Questions

Comments to the Author

1. Does the manuscript provide a valid rationale for the proposed study, with clearly identified and justified research questions?

Reviewer #1: Yes

2. Is the protocol technically sound and planned in a manner that will lead to a meaningful outcome and allow testing the stated hypotheses?

Reviewer #1: Yes

3. Is the methodology feasible and described in sufficient detail to allow the work to be replicable?

Reviewer #1: Yes

4. Have the authors described where all data underlying the findings will be made available when the study is complete?

Reviewer #1: Yes

5. Is the manuscript presented in an intelligible fashion and written in standard English?

Reviewer #1: Yes

6. Review Comments to the Author

You may also provide optional suggestions and comments to authors that they might find helpful in planning their study.

Reviewer #1:

1- My comments are primarily evaluation of the statistical design, data analysis and data management.

Author’s response: The authors thank the reviewer for the time dedicated to evaluating the manuscript. All comments have been carefully considered, and the manuscript has been revised accordingly.

2-The protocol is well written from a study design and analysis perspective. The sample size is derived appropriately with power considerations. The effect size of 0.14 is reasonable.

Author’s response: The authors thank the reviewer for the positive evaluation of the study design, statistical analysis, and sample size calculation.

3-However, the exact endpoint (presumably VAS if that is the primary) should be given and the appropriate measure being reported since this appears to be a repeated measures design. The Friedman test given in the ‘Statistical Analysis’ section is certainly appropriate.

Author’s response: The authors thank the reviewer for this important observation. The primary outcome has now been explicitly defined as pain intensity measured by the Visual Analog Scale (VAS). Lines: 286, 292, 315, 335, 341, 356, 374, 394, 399, 403, 410

Additionally, the repeated-measures nature of this outcome has been clarified in the manuscript, and the description of the statistical analysis has been refined accordingly (lines 448-450).

4-Figure 2 appears to be complete for timeline and assessments.

Author’s response: The authors thank the reviewer for this positive assessment of Figure 2.

5-There are some issues, easily addressed by the investigators.

The study needs a data management and data quality control section. These are missing and one does not see them in the supplemental materials.

Author’s response: The authors thank the reviewer for this important observation. A dedicated section on data management and data quality control has now been included in the manuscript to ensure transparency and methodological rigor (lines 465-470).

6- Also, are there going to be any issues with missing data, and if so, how will that be handled?

Author’s response: The authors thank the reviewer for this important observation. Given the short follow-up period, missing data are expected to be minimal. However, procedures have been implemented to monitor and report missing data, and appropriate statistical methods will be applied as needed. This information has been added to the manuscript (lines 459-463).

7. PLOS authors have the option to publish the peer review history of their article (what does this mean?). If published, this will include your full peer review and any attached files.

Do you want your identity to be public for this peer review? For information about this choice, including consent withdrawal, please see our Privacy Policy.

Reviewer #1: No

---

## [Decision Letter · Decision Letter 1]

23 Apr 2026

Photobiomodulation for pain management during placement of the copper T 380 intrauterine device: protocol for a randomized, double-blind controlled trial

PONE-D-26-05815R1

Dear Dr. Horliana,

We’re pleased to inform you that your manuscript has been judged scientifically suitable for publication and will be formally accepted for publication once it meets all outstanding technical requirements.

Kind regards,

Syed Khurram Azmat, PhD, MPH, MD

Academic Editor

PLOS One

Additional Editor Comments (optional):

Reviewers' comments:

Reviewer's Responses to Questions

**Comments to the Author**

1. Does the manuscript provide a valid rationale for the proposed study, with clearly identified and justified research questions?

Reviewer #1: Yes

2. Is the protocol technically sound and planned in a manner that will lead to a meaningful outcome and allow testing the stated hypotheses?

Reviewer #1: Yes

3. Is the methodology feasible and described in sufficient detail to allow the work to be replicable?

Reviewer #1: Yes

4. Have the authors described where all data underlying the findings will be made available when the study is complete?

Reviewer #1: Yes

5. Is the manuscript presented in an intelligible fashion and written in standard English?

Reviewer #1: Yes

6. Review Comments to the Author

You may also provide optional suggestions and comments to authors that they might find helpful in planning their study.

Reviewer #1: All comments have been addressed and the revisions noted in the manuscript.

XXXXXXXXXXXXXXXXXXXXXXXXXXXXXXXXXXXXXXXXXXXXXXXXXXXXXXXXXXXXXXXXXXXXXXXXXXXXXXXXXXXXXXXXXXXXXXXXXXXXXXXXXXXXXXXXXXXXXXXXXXXXXXXXXXXXXXXXXXXXXXXXXXXXXXXXXXX

7. PLOS authors have the option to publish the peer review history of their article (what does this mean?). If published, this will include your full peer review and any attached files.

Reviewer #1: No

---

## [Editor Report · Acceptance letter]

PONE-D-26-05815R1

PLOS One

Dear Dr. Horliana,

I'm pleased to inform you that your manuscript has been deemed suitable for publication in PLOS One. Congratulations! Your manuscript is now being handed over to our production team.

Kind regards,

on behalf of

Dr. Syed Khurram Azmat

Academic Editor

PLOS One